# The Impact of Coordinated Development of Ecological Environment and Technological Innovation on Green Economy: Evidence from China

**DOI:** 10.3390/ijerph19126994

**Published:** 2022-06-07

**Authors:** Lining Su

**Affiliations:** School of Management, Anhui University, Hefei 230601, China; sulining@vip.163.com; Tel.: +86-138-6678-6803

**Keywords:** ecological environment, technological innovation, coupling coordination, green economy

## Abstract

Promoting the coordinated development of ecological environment and technological innovation is significant to the development of a green economy. In this study, we construct an index system of ecological environment, technological innovation, and green economy based on the panel data of 30 provinces and cities in China from 2005 to 2016, using the entropy weight method, the coupling coordination model, and the panel vector autoregressive model (PVAR) to calculate the comprehensive development levels of ecological environment, technological innovation, and green economy and the coordination degree between ecological environment and technological innovation, and then further explore the impact of the coordinated development level of ecological environment and technological innovation on the development of a green economy. The research results include: First, from 2005 to 2016, the comprehensive development levels of ecological environment, technological innovation, and green economy in China’s 30 provinces and cities achieved different degrees of improvement as a whole. Among them, the comprehensive development level of green economy was the highest, followed by the development level of technological innovation, and the comprehensive development level of ecological environment was the lowest. Second, from 2005 to 2016, the coordination degree between ecological environment and technological innovation in China’s provinces and cities increased year by year, but on the whole, the coordination degree between ecological environment and technological innovation in various regions was in a state of imbalance. Third, there was a long-term equilibrium relationship among the coordinated development levels of ecological environment, technological innovation, and green economy. Fourth, through pulse analysis and Monte Carlo simulation, we found that the coordinated development level of ecological environment and technological innovation had a lagging positive impact on green economy. Finally, we provide a summary of the results of this study.

## 1. Introduction

Since the reform and opening up in 1978, China’s economy has grown rapidly with China becoming the largest developing country; China’s GDP exceeded 100 trillion yuan in 2020, second only to the United States in overall economic strength [1]. However, during the process of economic development, a serious environmental crisis has appeared in China in recent years; according to statistics, economic losses caused by environmental pollution account for 2–3% of the gross domestic product (GDP) every year. In addition, China’s economic growth rate is facing increasing downward pressure. In order to cope with the environmental crisis and enhance the power of economic development, China proposes to vigorously develop green economy (GE) and technological innovation (TI), in order to achieve sustainable economic and social development through vigorously developing a green economy and scientific and technological innovations. Technological innovation is the driving force for high-quality development of a green economy [2], and an ecological environment (EE) is an important guarantee for high-quality development of a green economy [3]. Protecting an ecological environment and promoting technological innovation play important driving roles in developing a green economy. However, it is still unclear how the coordination degree between ecological environment and technological innovation affect a green economy. Analyzing the impact of the coordination level between ecological environment and technological innovation on green economy is of practical significance for China’s high-quality development. In this study, we evaluate the comprehensive development levels of ecological environment, technology innovation, and green economy, and then evaluate the coordination degree between ecological environment and technological innovation and analysis the impact of coordination level between ecological environment and technological innovation on green economy. Finally, based on the research findings, we provide some policy suggestions.

The arrangement of this paper is as follows: First, we identify and describe the related research by searched and reading the literature related to the relationships among ecological environment, technological innovation, and green economy from the Web of Science database. Then, we introduce the research methods and data sources; explain the entropy method, the coupling coordination model, and the PVAR model; and design an index system of environmental, technology innovation, and green economy. Next, we display and discuss the results, which include assessing thecomprehensive development levels of environment, technological innovation and green economy; assessing the coordination degree between environment and technological innovation; and analyzing the impact of the coordination level between environment and technological innovation on green economy. Finally, we summarize the research findings.

## 2. Literature Review

At present, there are many studies on the relationship between ecological environment and economic development. Among them, Shi et al. (2019) measured and analyzed the coupling coordination and spatio-temporal heterogeneity of economic development and ecological environment in 17 tropical and subtropical regions from 2003 to 2016 by using a geographical and time weighted regression (GTWR) method, and found that there was significant spatio-temporal heterogeneity between economic development and ecological environment with most regions belonging to the economic lag type [4]. Zhu et al. (2021) took Guangdong Province of China as an example, and established a model of the coordination degree between ecological environment and economic development, then analyzed the coordination degree between ecological environment and economic development in Guangdong Province. They found that the coordination degree showed an increasingly high trend, which indicated that with the rapid economic development of Guangdong Province, more and more attention had been paid to environmental protection [5]. In addition, some scholars have used an improved entropy weight method to calculate the coupling and coordinated development of high-quality economic development and ecological environment in the Yellow River Basin of China from 2008 to 2018, and found that high-quality economic development, ecological environment quality, and their coupling and coordination level in the Yellow River Basin showed different degrees of improvement [6].

Regarding the relationship between technological innovation and economy, as early as the last century, some scholars had proven the role of technological innovation in promoting economic growth through research [7]. Later, scholars studied the specific impact of technological innovation on economic development. Among them, Liu, C. (2016) suggested that technological innovation was an important driving force for a country’s sustainable economic development and social progress, and technological innovation could be achieved through R&D investment, thereby achieving sustainable economic growth [8]. Wang et al. (2020) found that technological innovation could drive the transformation of the economic development mode, and economic development provided financial support for technological innovation, which showed coordination [9]. Some scholars have proposed to use technological innovation to promote the development of agricultural economy, and have pointed out that it was necessary to promote the role of technological innovation in the transformation of agricultural economic development from various aspects such as policy, investment, organizational construction, and the creation of an entrepreneurial environment [10].

Regarding the relationship between ecological environment and technological innovation, research on the relationship between them is also relatively early. For example, Frank (1997) and others proved that technology research was helpful to protect the ecological environment and put forward the concept of “ecological innovation” [11]. Chen (2010) and others explored the impact of technological innovation efficiency on the ecological environment by constructing a technological innovation index system and using data envelopment analysis (DEA) [12]. From the perspective of input and output, other scholars have used collected panel data to evaluate the green technology level and environmental governance performance of provincial enterprises in Anhui Province from multiple dimensions [13]. In addition, regarding the relationships among ecological environment, technological innovation, and green economy, most studies have focused on measuring the relationship between two of the three [14,15], while there are few studies on the relationships among the three. Among them, Zhao et al. (2021) used an entropy weight method, coupling coordination model, and gravity model to study the spatio-temporal coupling coordination relationship and spatio-temporal characteristics of the economy, energy, and ecological environment in the Yellow River Basin of China [16].

After reviewing the literature, we found that existing studies have mainly focused on analyzing the relationships between ecological environment and technological innovation, between ecological environment and economic development, as well as between technological innovation and economic development. Few studies in the literature have analyzed the relationships among ecological environment, technological innovation, and economic development together. Therefore, based on the panel data of 30 provinces and cities in China from 2005 to 2016, in this study, we construct an index system of ecological environment, technological innovation, and green economy by using the entropy weight method, a coupling coordination model, and a PVAR model to explore the impact of the coordinated development level of ecological environment and technological innovation on green economy, and finally put forward policy suggestions according to the research findings.

## 3. Research Methods and Data Sources

The purpose of this study is to analyze the impact of coordinated development of ecological environment and technological innovation on green economy. The research design mainly consisted of two parts. In the first part, we measure the level of coordinated development between ecological environment and technological innovation. In the second part, based on the calculation results of the coordination level of ecological environment and technological innovation, we analyze the impact of their coordination level on green economy. However, before the above analysis, it is necessary to evaluate the comprehensive development levels of ecological environment, technological innovation, and green economy separately. Therefore, the specific research design included three parts: First, it is necessary to evaluate the comprehensive development levels of ecological environment, technological innovation, and green economy. Then, the coordination level of ecological environment and technological innovation needs to be calculated. Finally, the impact of the coordination level between ecological environment and technological innovation on green economy is analyzed.

### 3.1. Index System for Ecological Environment, Technological Innovation, and Green Economy

Comprehensive development level is a description of the development state of a thing. At present, to evaluate the comprehensive development levels of ecological environment, technological innovation, and green economy, scholars have either used a single index to measure, or they have designed an index system, and then used factor analysis, the analytic hierarchy process, and other methods to calculate the data in the index system, and finally, evaluated the development levels of ecological environment, technological innovation, and green economy. In this study, an index system is designed to evaluate the comprehensive development levels of ecological environment, technological innovation, and green economy. This is mainly because an index system can comprehensively investigate the state of ecological environment, technological innovation, and green economy.

In terms of ecological environment, referring to the existing studies [17,18], in this study, we decompose the ecological environment into three first-level indicators: environmental pollution generation, environmental pollution control, and natural environment foundation. We build eleven secondary-level indicators, which include industrial solid waste generation (ten thousand tons), total wastewater discharge (ten thousand tons), total sulfur dioxide discharge (ten thousand tons), industrial solid waste investment (100 million yuan), industrial wastewater treatment investment (100 million yuan), wetland area (ten thousand hm^2^), and forest coverage (%), etc. (see Table 2). In terms of technological innovation, based on the existing research [19,20,21], we divide technological innovation into three first-level indicators, which include investment in scientific and technological innovation, the output of technological innovation, and technological innovation effectiveness, and secondary-level indicators which include the proportion of enterprises that include Research and Experimental Development (R&D) institutions (%), the full-time equivalent of ten thousand R&D personnel, the number of scientific papers (article), the number of patent authorizations (items), the sales revenue of new products (ten thousand yuan) and the export value of high-tech products (100 million yuan), etc. (see Table 2). In terms of green economy, based on existing studies [22,23], we divide economic growth into three first-level indicators, which include economic growth scale, economic growth structure, and economic growth benefits [22,23]. Based on first-level indicators, we build thirteen secondary-level indicators, including GDP (100 million yuan), industrial production value (100 million yuan), tertiary industry added value as a percentage of GDP (%), retail sales of consumer goods per capita (yuan), income gap (yuan), urbanization rate (%), etc. (see Table 1).

### 3.2. Comprehensive Evaluation of Ecological Environment, Technological Innovation, and Green Economy

In this study, it is necessary to comprehensively evaluate the comprehensive development levels of ecological environment, technological innovation, and green economy. We chose the entropy method to evaluate the comprehensive development levels of environment, technological innovation, and green economy. The purpose of the entropy method is to objectively determine the weight of indicators through the dispersion degree of indicators in the samples. Generally speaking, the smaller the information entropy of an indicator is, the higher the degree of aggregation of the indicator, and the greater the role of the indicator on behalf of the whole indicator system. Therefore, the weight of the indicator is also higher. The specific process is as follows:

First, the original data need to be standardized to eliminate the impact of different measurement units and dimensions on the indicators. The calculation formulas are as follows:(1)$$\mathrm{Positive}\;\mathrm{indicator}:\;y_{kij}=\frac{x_{ij}-\min\left(x_{ij}\right)}{\max\left(x_{ij}\right)-\min\left(x_{ij}\right)}$$
(2)$$\mathrm{Negative}\;\mathrm{indicator}:\;\;y_{kij}=\frac{\max\left(x_{ij}\right)-x_{ij}}{\max\left(x_{ij}\right)-\min\left(x_{ij}\right)}$$
where *x_ij_* represents sample value; max (*_kij_*) and min (*x_ij_*), respectively, represent maximum and minimum values in the sample data.

Second, the entropy method in the objective weighting method is used to calculate the weight of each index as follows:(3)$$p_{kij}=\frac{y_{ij}}{\sum_{i=1}^{m}y_{ij}}\;;\;e_{kj}=\left[-\frac{1}{ln\left(m\right)}\right]\overset{m}{\underset{i=1}{\sum }}p_{ij}lnp_{ij}$$
(4)$$w_{j}=\left(1-e_{i}\right)/\overset{m}{\underset{i=1}{\sum }}\left(1-e_{i}\right)$$

Finally, the comprehensive development levels of ecological environment (*U_E_*), technological innovation (*U_T_*), and green economy *(U_G_*) are calculated. The specific calculation formulas are as follows:(5)$$U=\overset{m}{\underset{i=1}{\sum }}w_{ij}y_{ij}\;;\;\overset{m}{\underset{i=1}{\sum }}w_{ij}=1$$

### 3.3. Coupling Coordination Model

Coupling originally comes from physics and refers to the interaction between two or more objects. At present, the coupling coordination degree model is widely used to analyze the coordination development level between different systems [24]. In this research, we analyze the coordinated development correlation between ecological environment and technological innovation through the coupling coordination model. The specific calculation formulas are as follows:(6)$$C=2\sqrt{\frac{U_{E}\times U_{T}}{{\left(U_{E}+U_{T}\right)}^{2}}},C\in \left[0,\;1\right]$$
(7)$$T=\alpha U_{E}+\beta U_{T},\alpha +\beta =1$$
(8)$$D=\sqrt{C\times T},D\in \left[0,\;1\right]$$

In Equation (6), *U_E_* is ecological environment, *U_T_* is technological innovation, and *C* is system coupling degree which can reflect consistent development between ecological environment and technological innovation. The larger *C* is, the higher the coupling and the better the synergy effect between ecological environment and technological innovation. When *C* = 0, the coupling degree between ecological environment and technological innovation reaches its lowest, indicating that the two systems are independent. When *C* = 1, the coupling degree between ecological environment and technological innovation reaches its peak, which shows that the two systems are in an orderly development state. However, a higher coupling degree can also be obtained when ecological environment and technological innovation are at a low level. Therefore, the coordination degree model is established based on the coupling degree model to explore the coordinated development level between ecological environment and technological innovation, as shown in Equations (7) and (8).

In Equations (7) and (8), T denotes a comprehensive evaluation index that can reflect the overall synergistic effect of ecological environment and technological innovation. It also reflects the impact of comprehensive development level on cooperative dispatching, which are undetermined coefficients. In this study, we assumed *α* = *β* = 0.5; *D* is coordination degree. The greater the value of *D*, the higher the degree of coordination between ecological environment and technological innovation, and the better coordinated development level of the two systems.

In this study, we divide the coordinated development level of ecological environment and technological innovation by referring to the research of previous scholars [25,26] (see Table 2).

### 3.4. Panel Vector Autoregressive Model

Sims (1980) established a single dimensional vector autoregressive model (VAR) to describe the impact of variables on a specific variable [27]. A characteristic of this model is that all variables are regarded as endogenous variables, that is, each endogenous variable is taken as a function of the lag value of all endogenous variables in the system to construct the model, and therefore, truly reflects the interaction between each variable. However, the VAR model does not support panel data. Therefore, later, Holtz Eakin, Newey, and Rosen (1988) extended it to panel data structure (PVAR) [28]. PVAR considers the individual and time fixed effects, increases the precision of estimation, and any and no presuppositions are made about the relationship among variables. In particular, no assumptions are made about the direction of mutual causation among variables. Therefore, in this study, we used the PAVR model to analyze the impact of the coordination level between ecological environment and technological innovation on green economy [29]. The PVAR model is as follow:(9)$$Y_{it}=\beta_{0}+\alpha_{i}+\overset{p}{\underset{j=1}{\sum }}\beta_{j}Y_{t-j}+\varepsilon_{it}\;(i=1,2,3,\ldots \ldots ,30\;,\;t=1,2,3\ldots \ldots ,12)$$
where *i* represents province; *t* represents year; *Y_it_* includes two column vectors, namely coordinated development level (*CO*) between ecological environment and technological innovation and green economy (*EC*); *β*_0_ represents intercept term; *p* represents lag order, *β_j_* represents parameter matrix of lag *j* order; *α_i_* represents variable of individual fixed effects; and *ε_it_* represents random disturbance term.

### 3.5. Data Sources

The data used in this study were derived from the China Statistical Yearbook (2006–2017) [30], the China Science and Technology Statistical Yearbook (2006–2017) [31], the China High Technology Statistical Yearbook (2006–2017) [32], the China Demographic and Employment Statistical Yearbook (2006–2017), and the China Economic and Social Data Research Platform [33,34]. Some data also came from the website of the National Bureau of Statistics and local statistical bulletins [35].

## 4. Results

### 4.1. The Comprehensive Development Levels of Ecological Environment, Technological Innovation, and Green Economy

The comprehensive calculation model is used to calculate the comprehensive development levels of ecological environment, technological innovation, and green economy in 30 provinces and cities. The results are shown in Table 3, Table 4 and Table 5 and Figure 1.

Combining Table 3, Table 4 and Table 5, it can be seen that from 2005 to 2016, the comprehensive development levels of ecological environment, technological innovation, and green economy in China achieved various degrees of improvement as a whole. The average annual comprehensive development level of ecological environment increased from 0.09 to 0.12, increasing by 133.3%. The development level of technological innovation increased from 0.03 to 0.17, with a huge growth rate of 566.7%. The average annual comprehensive development level of green economy increased from 0.16 to 0.82, and the growth rate also reached an alarming 512.5%. It can be seen that from 2005 to 2016, the comprehensive development level of the green economy improved the most, the development level of and technological innovation was second, and the comprehensive development level of the ecological environment improved the least. This shows that China has indeed achieved great success in transforming its development model and moving towards high-quality development. China’s green economy development trend has achieved good results. China has always paid attention to the country’s technological innovation capabilities, and has achieved good results by continuously supporting the development of high-tech industries. Although the development of the ecological environment has also been improved, it has not achieved the obvious effect, China still needs to invest more in ecological and environmental protection.

As can be seen from Figure 1, the comprehensive development levels of China’s ecological environment, technological innovation, and green economy generally shows a continuous development trend from 2005 to 2016. The development curve of green economy shows a steady upward trend year by year, the comprehensive curve of technological innovation shows a slow rise at first, then a slight decline, and then a slow upward trend, showing a slow rise as a whole. Specifically, it gradually increased slightly from 2005 to 2010, began to decline after reaching a small peak in 2010, and gradually increased two years later from 2012 to 2016. The ecological environment curve shows a trend of first falling and then rising, and generally shows a slow development trend. Specifically, it continued to decline from 2009 to 2012, and the lowest point in 2012 was 0.05, and then slowly fluctuated until 2016, with the annual average rising from 0.09 to 0.12. It can be found that the fluctuation trends of the technological innovation curve and the ecological environment curve are relatively close, and the two intersected once in 2009. The two are quite different from the green economic curve, which shows a rapid upward trend. This shows that, as compared with the rapid development of China’s green economy, the development trend of technological innovation and ecological environment is not satisfactory. Although China’s technological innovation has made progress in recent years, there are still “bottlenecks” in high-tech industries, and many cutting-edge technologies have not been broken through. With the rapid development of China’s economy, all types of environmental pollution have also followed, resulting in significant harm to the ecological environment. Even if the Chinese government advocates vigorously developing green economy and environmental protection, the actual results are not good.

### 4.2. Coordination Degree between Ecological Environment and Technological Innovation

By using the coupling coordination model, the coordination degree between ecological environment and technological innovation is calculated as described below (see Table 6).

It can be seen from Table 6 that coordination degrees between ecological environment and technological innovation increase year by year from 2005 to 2016, but on the whole, the coordination degrees between ecological environment and technological innovation in various regions were in a state of imbalance. Taking 2005, 2010, and 2016 as examples, in 2005,coordination degrees between ecological environment and technological innovation in China’s provinces and cities ranged from 0.09 to 0.21, and most were in a seriously unbalanced state. By 2010, as compared with 2005, the coordination degrees between ecological environment and technological innovation in China’s provinces cities had slightly improved, ranging from 0.14 to 0.22, and were still in a serious state of imbalance. By 2016, the coordination degrees between ecological environment and technological innovation in China’s provinces and cities further improved, ranging from 0.19 to 0.36, and were in a state of moderate imbalance. It can be seen that although the coordination degrees between ecological environment and technological innovation in China’s provinces and cities show an upward trend year by year, coordination levels have always been in a state of serious imbalance.

### 4.3. Analysis of the Impact of Coordination Degree between Ecological Environment and Technological Innovation on Green Economy

#### 4.3.1. Unit Root Test

First, a unit root is used to test stationary; it examines whether a variable is in the same order and a single integer or not, to avoid deviation of estimation results from the existence of pseudo regression. It can be seen from Table 7 that the original data of the coordinated development level (CO) is not stable. In order to achieve a single integration of the same order, we performed first-order difference processing on the coordinated development level (EC) data. The results show that the green economy data of all regions are stable after the first-order difference, and the co-integration test can be carried out. Similarly, the unit root test of green economy data shows that the original data and the first-order difference data are stable. It can be seen that the coordinated development degree and green economy are stable after the first-order difference.

#### 4.3.2. Co-Integration Test

According to the unit root test, there is a single integration of the same order between the coordinated development level and green economy. Therefore, a co-integration test can be performed to test whether or not a long-term equilibrium relationship exists between the two. According to the co-integration test method of Pedroni, we tested two variables. The results are shown in Table 8. According to the results of the Pedroni co-integration test, the results are not significant, based on the statistics of Panel v, Panel rho, and Group PP, while they are significant at the 1% level in the Group ADF statistics. and they are significant at the 0.1% level in the statistics of Panel ADF, Panel PP, and Group rho. According to the results of the Johansen co-integration test, there is at least one co-integration relationship. In conclusion, there is a long-term equilibrium relationship between the level of coordinated development and the green economy.

#### 4.3.3. Optimal Lag Order Test

It is necessary to discover optimal lag order to test the panel vector autoregressive model. The test results of the maximum lag order of the PVAR model are shown in Table 9. Among them, AIC, BIC, and HQIC show that the PVAR model with a lag order of two should be constructed.

#### 4.3.4. Granger Causality Test

After the co-integration relationship test, it is found that there is a long-term equilibrium relationship between coordinated development level (CO) and green economy (GE). Therefore, the Granger causality test can be performed to explore a causal relationship [30,31]. The specific test results are shown in Table 10. According to the test results, the original hypothesis that green economy is not the cause of an increase in coordinated development level is rejected at the significance level of 0.1% between coordinated development level (CO) and green economy (GE), and the original hypothesis that coordinated development level is not the cause of green economy is rejected at the significance level of 0.1%. This shows that there is a Granger causality relationship between coordinated development level (CO) and green economy (GE) after the first-order difference at the significance level of 0.1%.

#### 4.3.5. Pulse Response Analysis

In this study, we used the impulse response function and the Monte Carlo simulation method to study the impact of the coordination of ecological environment and technological innovation on green economy. In the simulation process, the number of periods set by the model is 15 and the number of repetitions of data simulation is 500; the impulse response function results are shown in Figure 2. The abscissa represents the number of lag periods; the ordinate represents the corresponding degree; the two curves at the top and bottom represent the upper and lower bounds of 95% confidence interval, respectively; and the middle curve represents the point estimation value of impulse response function.

Figure 2 represents the impulse response results of the coordinated development level and green economy. Figure 2a,d, respectively, show the response of coordinated development level and green economy to the impact of one standard deviation unit; both reach their peak in the current period and show a maximum positive response. Then, the impact gradually weakens and approaches zero response in Phase 1. Then, the green economy has a negative impact on the coordinated development level, and the negative impact gradually increases and reaches the maximum in Phase 2. Since then, the negative impact gradually weakens and approaches zero. Figure 2b shows the dynamic impact of green economy on the level of coordinated development. It can be seen from the Figure 2 that the coordinated development level has a significant negative response to green economy in the current period and reaches the maximum negative response, but then the negative impact of green economy on the coordinated development level gradually weakens. In phase 2, the impact of green economy on the coordinated development level is close to zero, then, presents a positive impact and reaches the maximum positive impact in phase 3, and then, the impact gradually weakens, close to zero. Figure 2c shows the dynamic relationship between the level of coordinated development and the impact of green economy. After the impact of a standard deviation of the current green economy, the initial response of the level of coordinated development is zero, but then there is a positive response, which gradually increases, reaches a peak in phase 2, and then gradually approaches zero, indicating that the level of coordinated development has a lagging positive impact on green economy.

## 5. Conclusions and Prospect

### 5.1. Conclusions

Based on the panel data of 30 provinces and cities in China from 2005 to 2016, in this study, we adopt the entropy weight method, coupling coordination model, and PAVR model to analyze the impact of coordinated development of ecological environment and technological innovation on green economy. The research finding include: First, from 2005 to 2016, the level of ecological environment, technological innovation, and green economy in 30 provinces and cities in China has achieved varying degrees of improvement. Among them, the level of green economy is the highest, followed by technological innovation, and the level of ecological environment is the lowest. This is mainly because China has increased economic stimulus and investment in technological innovation and the Chinese government attaches more importance to the economy and technology than to environmental protection. Second, the levels of coordination degree between ecological environment and technological innovation increased year by year from 2005 to 2016, but on the whole, the degree of coordination between ecological environment and technological innovation in different regions was unbalanced. Thirdly, the coordination degree between ecological environment and technological innovation has a backward positive impact on green economy, that is, coordinated development between ecological environment and technological innovation has a positive impact on the development of green economy, but the coordination level of the two is low, and therefore, it plays a small role in promoting the development of a green economy. This shows that China needs to promote the development of a green economy by promoting technological innovation and environmental protection. Based on the above results, it can be found that improving the coordination level of technology and ecological environment can promote the development of green economy, but China’s work on ecological environment is still insufficient as compared with technological innovation and the development of green economy.

### 5.2. Research Limitation and Prospect

Based on China’s provincial panel data, we conducted an empirical analysis by using the coupling coordination model and panel vector autoregressive model. Our results show that the coordinated development level of China’s ecological environment and technological innovation is increasingly important to the development of green economy, which provides theoretical support for the sustainable and healthy development of a green economy. However, there are still some shortcomings. First, in this study, we did not analyze various regions in China, and the research time was only ten years. In the future, if conditions permit, China could be divided into specific regions, and research could be carried out according to different regional characteristics to explore the impact of the coordinated development level of ecological environment and technological innovation in different regions on the development of green economy. At the same time, the research period could be appropriately increased. Second, econometric models are diverse and complex. In this study, we only used the coupling coordination model and panel vector autoregressive model. Therefore, using other models to study this topic should become a research direction in the future.

## Figures and Tables

**Figure 1 ijerph-19-06994-f001:**
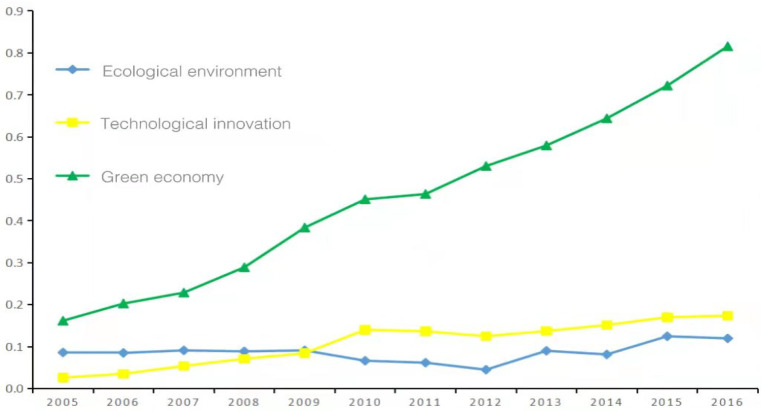
Comprehensive development levels of EE, TI, and GE (2005–2016).

**Figure 2 ijerph-19-06994-f002:**
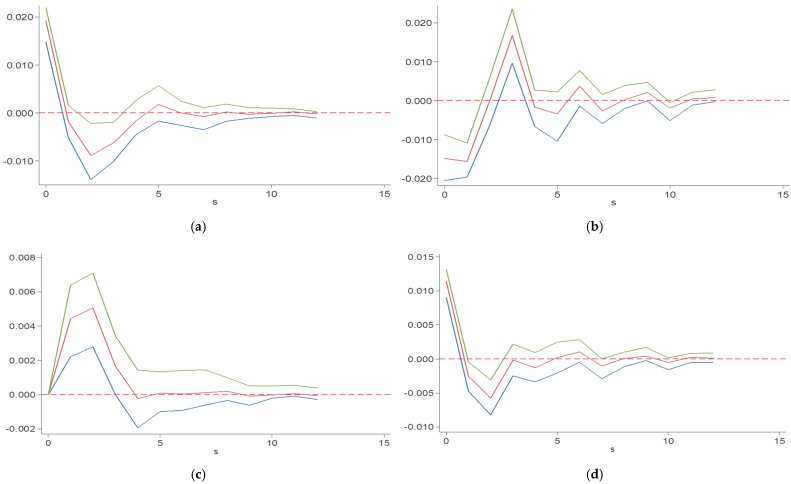
Corresponding results of pulses: (**a**) GE response to GE shock; (**b**) GE response to CO shock; (**c**) CO response to GE shock; (**d**) CO response to CO shock.

**Table 1 ijerph-19-06994-t001:** Index system for EE, TI, and GE.

Total System	First Indicator	Secondary Indicator
Ecological environment	Environmental pollution generation	Industrial solid waste generation (ten thousand tons), total wastewater discharge (ten thousand tons), total sulfur dioxide discharge (ten thousand tons)
	Environmental pollution control	Completion of investment in forestry fixed assets (100 million yuan), investment in industrial waste gas treatment (100 million yuan), industrial solid waste investment (100 million yuan), industrial wastewater treatment investment (100 million yuan)
	Natural environment foundation	Wetland area (ten thousand hm^2^), area of nature reserve (thousand hm^2^), total groundwater resources (a hundred million m^3^), forest coverage (%)
Technological innovation	Technological innovation investment	The full-time equivalent of R&D personnel per 10,000 people, the proportion of R&D expenditure in GDP (%), the proportion of enterprises with R&D institutions (%), the proportion of R&D expenditure in main business income (%)
	Scientific and technological innovation output	Number of scientific papers (article), number of patents (items), technology market turnover (100 million yuan)
	Scientific and technological innovation effectiveness	Sales revenue of new products (ten thousand yuan), export value of high-tech products (100 million yuan), energy consumption per unit GDP (ton of standard coal/ten thousand yuan), labor productivity (ten thousand yuan/person)
Economic growth	Economic growth scale	GDP (100 million yuan), industrial production value (100 million yuan), social fixed asset investment (100 million yuan), financial development level, fiscal revenue (ten thousand yuan)
	Economic growth structure	The proportion of the added value of the tertiary industry in GDP (%), the proportion of the added value of the secondary industry in GDP (%), and the retail sales of consumer goods per capita (yuan)
	Economic growth benefits	Per capita GDP (yuan), income gap (Yuan), urbanization rate (%), kilometer density (km/10,000 square meters), total savings of urban and rural residents (100 million yuan),per capita GDP (yuan), income gap (Yuan), urbanization rate (%), kilometer density (km/10,000 square meters), total savings of urban and rural residents (100 million yuan)

**Table 2 ijerph-19-06994-t002:** Classification of coordination degree.

Stage	Antagonistic Stage				Run-In Stage			Coordination Stage		
Coordination degree	0 ≤ D < 0.4				0.4 ≤ D < 0.8			0.8 ≤ D < 1.0		
D	0 ≤ D < 0.1	0 ≤ D < 0.2	0.2 ≤ D < 0.3	0.3 ≤ D < 0.4	0.4 ≤ D < 0.5	0.5 ≤ D < 0.6	0.6 ≤ D < 0.7	0.7 ≤ D < 0.8	0.8 ≤ D < 0.9	0.9 ≤ D≤ 1.0
Classification	Extreme imbalance	Serious disorder	Moderate Disorder	Mild disorder	On the verge of maladjustment	Barely coordinated	Primary coordination	Intermediatecoordination	Good coordination	Quality coordination

**Table 3 ijerph-19-06994-t003:** The comprehensive development levels of ecological environment.

Province	2005	2006	2007	2008	2009	2010	2011	2012	2013	2014	2015	2016
Beijing	0.04	0.06	0.06	0.05	0.03	0.02	0.03	0.03	0.07	0.04	0.07	0.07
Tianjin	0.04	0.03	0.03	0.03	0.03	0.02	0.05	0.01	0.04	0.05	0.06	0.03
Hebei	0.14	0.05	0.05	0.06	0.1	0.04	0.05	0.03	0.04	0.05	0.05	0.04
Shanxi	0.08	0.15	0.15	0.18	0.13	0.11	0.08	0.06	0.08	0.04	0.06	0.04
Inner Mongolia	0.09	0.12	0.13	0.13	0.14	0.09	0.11	0.08	0.28	0.18	0.2	0.2
Liaoning	0.06	0.06	0.07	0.06	0.05	0.04	0.03	0.02	0.07	0.05	0.13	0.07
Jilin	0.09	0.07	0.08	0.08	0.08	0.08	0.05	0.03	0.04	0.04	0.06	0.06
Heilongjiang	0.07	0.08	0.11	0.09	0.15	0.11	0.08	0.04	0.14	0.11	0.18	0.17
Shanghai	0.01	0.05	0.01	0.02	0.02	0.01	0.01	0.01	0.02	0.04	0.05	0.1
Jiangsu	0.09	0.08	0.12	0.09	0.08	0.03	0.06	0.04	0.13	0.06	0.14	0.15
Zhejiang	0.04	0.06	0.06	0.05	0.06	0.03	0.03	0.03	0.09	0.1	0.19	0.18
Anhui	0.03	0.04	0.05	0.04	0.04	0.03	0.02	0.02	0.05	0.04	0.07	0.17
Fujian	0.19	0.1	0.05	0.05	0.04	0.04	0.03	0.03	0.08	0.07	0.2	0.19
Jiangxi	0.09	0.08	0.08	0.08	0.13	0.08	0.06	0.03	0.06	0.05	0.11	0.11
Shandong	0.2	0.13	0.14	0.23	0.12	0.1	0.16	0.09	0.06	0.07	0.19	0.16
Henan	0.13	0.09	0.12	0.12	0.05	0.05	0.03	0.01	0.07	0.04	0.07	0.07
Hubei	0.05	0.05	0.06	0.05	0.08	0.06	0.02	0.02	0.06	0.06	0.09	0.32
Hunan	0.11	0.13	0.1	0.1	0.12	0.12	0.08	0.19	0.08	0.07	0.11	0.11
Guangdong	0.15	0.13	0.12	0.08	0.07	0.06	0.05	0.06	0.08	0.07	0.19	0.13
Guangxi	0.08	0.06	0.12	0.1	0.11	0.18	0.14	0.12	0.3	0.32	0.33	0.18
Hainan	0.05	0.06	0.05	0.05	0.07	0.03	0.05	0.02	0.05	0.06	0.06	0.07
Chongqing	0.05	0.04	0.07	0.08	0.06	0.03	0.04	0.01	0.05	0.04	0.08	0.06
Sichuan	0.16	0.15	0.19	0.18	0.2	0.06	0.09	0.05	0.12	0.12	0.17	0.16
Guizhou	0.05	0.09	0.07	0.09	0.07	0.05	0.08	0.02	0.05	0.05	0.07	0.06
Yunnan	0.1	0.11	0.12	0.11	0.07	0.07	0.04	0.03	0.08	0.11	0.14	0.09
Shaanxi	0.04	0.03	0.06	0.04	0.09	0.07	0.07	0.04	0.07	0.05	0.08	0.07
Gansu	0.08	0.11	0.12	0.07	0.13	0.08	0.05	0.07	0.07	0.07	0.07	0.1
Qinghai	0.12	0.14	0.14	0.15	0.2	0.17	0.15	0.08	0.18	0.21	0.3	0.21
Ningxia	0.03	0.05	0.05	0.05	0.05	0.02	0.02	0.02	0.03	0.06	0.04	0.05
Xinjiang	0.12	0.15	0.15	0.15	0.16	0.11	0.09	0.06	0.16	0.12	0.18	0.17
Average	0.09	0.09	0.09	0.09	0.09	0.07	0.06	0.05	0.09	0.08	0.12	0.12

**Table 4 ijerph-19-06994-t004:** The comprehensive development levels of technological innovation.

Province	2005	2006	2007	2008	2009	2010	2011	2012	2013	2014	2015	2016
Beijing	0.08	0.11	0.18	0.23	0.31	0.42	0.42	0.46	0.47	0.52	0.59	0.58
Tianjin	0.06	0.07	0.1	0.14	0.14	0.23	0.23	0.21	0.26	0.29	0.33	0.34
Hebei	0.01	0.02	0.03	0.04	0.05	0.09	0.09	0.08	0.09	0.1	0.11	0.11
Shanxi	0.01	0.02	0.03	0.04	0.04	0.09	0.08	0.07	0.08	0.09	0.1	0.1
Inner Mongolia	0.02	0.02	0.03	0.03	0.03	0.08	0.07	0.05	0.06	0.06	0.07	0.08
Liaoning	0.02	0.03	0.06	0.07	0.09	0.12	0.12	0.12	0.13	0.15	0.17	0.18
Jilin	0.01	0.02	0.03	0.03	0.04	0.06	0.06	0.06	0.06	0.07	0.07	0.08
Heilongjiang	0.01	0.02	0.03	0.03	0.05	0.07	0.07	0.07	0.07	0.08	0.09	0.09
Shanghai	0.05	0.06	0.11	0.16	0.21	0.32	0.32	0.31	0.34	0.37	0.41	0.42
Jiangsu	0.09	0.11	0.15	0.22	0.27	0.37	0.37	0.35	0.4	0.44	0.49	0.49
Zhejiang	0.07	0.08	0.12	0.16	0.17	0.29	0.29	0.25	0.28	0.31	0.35	0.36
Anhui	0.03	0.04	0.05	0.07	0.08	0.16	0.16	0.13	0.14	0.15	0.17	0.17
Fujian	0.02	0.03	0.06	0.07	0.09	0.16	0.15	0.13	0.17	0.18	0.21	0.22
Jiangxi	0.01	0.02	0.03	0.04	0.04	0.07	0.06	0.06	0.07	0.08	0.09	0.09
Shandong	0.03	0.04	0.07	0.1	0.11	0.17	0.17	0.16	0.18	0.2	0.22	0.23
Henan	0.01	0.02	0.03	0.04	0.05	0.09	0.08	0.07	0.07	0.08	0.09	0.09
Hubei	0.04	0.05	0.06	0.08	0.08	0.12	0.12	0.13	0.13	0.14	0.16	0.16
Hunan	0.02	0.03	0.04	0.06	0.07	0.12	0.11	0.1	0.1	0.11	0.12	0.13
Guangdong	0.06	0.08	0.12	0.17	0.22	0.3	0.3	0.28	0.36	0.4	0.43	0.44
Guangxi	0.01	0.01	0.01	0.02	0.02	0.03	0.03	0.03	0.03	0.04	0.04	0.04
Hainan	0.01	0.01	0.02	0.02	0.03	0.05	0.05	0.04	0.06	0.06	0.07	0.07
Chongqing	0.02	0.03	0.05	0.07	0.07	0.14	0.13	0.11	0.12	0.13	0.16	0.16
Sichuan	0.01	0.02	0.03	0.03	0.04	0.11	0.11	0.08	0.05	0.06	0.07	0.07
Guizhou	0.01	0.01	0.01	0.01	0.01	0.04	0.04	0.03	0.03	0.03	0.03	0.03
Yunnan	0.01	0.02	0.02	0.03	0.03	0.06	0.06	0.05	0.06	0.06	0.07	0.07
Shaanxi	0.02	0.03	0.05	0.06	0.07	0.15	0.14	0.13	0.1	0.11	0.13	0.13
Gansu	0.01	0.01	0.02	0.03	0.03	0.06	0.06	0.05	0.04	0.05	0.06	0.06
Qinghai	0.01	0.01	0.02	0.02	0.02	0.06	0.05	0.03	0.04	0.05	0.05	0.05
Ningxia	0.01	0.02	0.03	0.04	0.04	0.1	0.09	0.06	0.07	0.08	0.09	0.1
Xinjiang	0.01	0.01	0.02	0.02	0.02	0.07	0.06	0.04	0.05	0.05	0.06	0.06
Average	0.026	0.04	0.05	0.07	0.08	0.14	0.14	0.12	0.14	0.15	0.17	0.17

**Table 5 ijerph-19-06994-t005:** The comprehensive development levels of green economy.

Province	2005	2006	2007	2008	2009	2010	2011	2012	2013	2014	2015	2016
Beijing	0.3	0.26	0.28	0.34	0.48	0.52	0.47	0.52	0.56	0.62	0.71	0.84
Tianjin	0.2	0.24	0.32	0.36	0.41	0.49	0.49	0.54	0.59	0.63	0.76	0.84
Hebei	0.14	0.19	0.23	0.29	0.38	0.45	0.49	0.55	0.6	0.6	0.74	0.85
Shanxi	0.17	0.21	0.23	0.28	0.4	0.45	0.47	0.55	0.51	0.65	0.75	0.83
Inner Mongolia	0.16	0.2	0.23	0.3	0.41	0.48	0.53	0.59	0.66	0.66	0.72	0.79
Liaoning	0.18	0.22	0.25	0.33	0.41	0.48	0.51	0.58	0.64	0.66	0.66	0.67
Jilin	0.17	0.23	0.25	0.28	0.38	0.43	0.42	0.56	0.56	0.64	0.75	0.86
Heilongjiang	0.18	0.21	0.21	0.28	0.37	0.44	0.47	0.54	0.59	0.66	0.73	0.84
Shanghai	0.17	0.23	0.3	0.37	0.44	0.51	0.49	0.51	0.52	0.56	0.64	0.84
Jiangsu	0.17	0.22	0.25	0.31	0.41	0.48	0.48	0.53	0.59	0.66	0.72	0.78
Zhejiang	0.16	0.22	0.27	0.37	0.45	0.53	0.53	0.56	0.66	0.68	0.69	0.79
Anhui	0.16	0.21	0.23	0.3	0.39	0.45	0.44	0.5	0.55	0.65	0.76	0.89
Fujian	0.12	0.17	0.19	0.28	0.4	0.43	0.46	0.52	0.59	0.69	0.76	0.82
Jiangxi	0.13	0.16	0.18	0.23	0.31	0.39	0.41	0.46	0.52	0.59	0.68	0.86
Shandong	0.18	0.22	0.24	0.29	0.38	0.45	0.47	0.54	0.58	0.65	0.73	0.84
Henan	0.19	0.23	0.26	0.29	0.37	0.42	0.4	0.51	0.58	0.65	0.73	0.84
Hubei	0.14	0.18	0.19	0.27	0.39	0.45	0.45	0.5	0.57	0.65	0.75	0.87
Hunan	0.13	0.16	0.2	0.28	0.38	0.44	0.45	0.49	0.56	0.64	0.75	0.88
Guangdong	0.17	0.22	0.27	0.33	0.4	0.44	0.47	0.53	0.58	0.65	0.7	0.84
Guangxi	0.18	0.2	0.21	0.26	0.35	0.42	0.44	0.51	0.59	0.67	0.78	0.89
Hainan	0.13	0.18	0.22	0.27	0.33	0.44	0.49	0.56	0.61	0.69	0.78	0.84
Chongqing	0.13	0.19	0.22	0.31	0.39	0.46	0.48	0.54	0.59	0.68	0.76	0.84
Sichuan	0.13	0.16	0.18	0.26	0.36	0.44	0.49	0.54	0.62	0.68	0.75	0.77
Guizhou	0.14	0.17	0.17	0.21	0.34	0.43	0.36	0.58	0.46	0.51	0.56	0.65
Yunnan	0.11	0.22	0.17	0.21	0.31	0.4	0.53	0.43	0.47	0.55	0.63	0.76
Shaanxi	0.19	0.21	0.24	0.31	0.38	0.45	0.5	0.55	0.62	0.69	0.78	0.74
Gansu	0.15	0.19	0.21	0.27	0.42	0.5	0.47	0.55	0.63	0.69	0.76	0.7
Qinghai	0.15	0.18	0.2	0.23	0.39	0.47	0.44	0.51	0.57	0.64	0.71	0.81
Ningxia	0.14	0.18	0.2	0.22	0.32	0.38	0.41	0.47	0.54	0.59	0.64	0.86
Xinjiang	0.18	0.22	0.26	0.34	0.36	0.41	0.4	0.59	0.67	0.74	0.78	0.84
Average	0.16	0.20	0.23	0.29	0.38	0.45	0.46	0.53	0.58	0.64	0.72	0.82

**Table 6 ijerph-19-06994-t006:** Coordination degree between ecological environment and technological innovation.

Province	2005	2006	2007	2008	2009	2010	2011	2012	2013	2014	2015	2016
Beijing	0.16	0.2	0.22	0.23	0.22	0.22	0.24	0.24	0.3	0.27	0.32	0.32
Tianjin	0.16	0.16	0.17	0.19	0.17	0.18	0.23	0.14	0.23	0.24	0.26	0.21
Hebei	0.15	0.13	0.14	0.16	0.19	0.17	0.18	0.15	0.17	0.18	0.19	0.18
Shanxi	0.13	0.16	0.19	0.2	0.19	0.22	0.2	0.18	0.2	0.17	0.2	0.18
Inner Mongolia	0.14	0.15	0.17	0.18	0.17	0.2	0.21	0.18	0.25	0.23	0.25	0.25
Liaoning	0.14	0.14	0.18	0.17	0.18	0.19	0.18	0.15	0.22	0.21	0.27	0.23
Jilin	0.13	0.13	0.15	0.15	0.16	0.18	0.16	0.14	0.16	0.16	0.18	0.19
Heilongjiang	0.12	0.13	0.17	0.17	0.21	0.21	0.19	0.16	0.22	0.22	0.25	0.25
Shanghai	0.11	0.17	0.14	0.17	0.18	0.16	0.17	0.16	0.19	0.25	0.27	0.32
Jiangsu	0.21	0.22	0.26	0.27	0.27	0.24	0.28	0.25	0.34	0.28	0.36	0.37
Zhejiang	0.17	0.19	0.2	0.21	0.23	0.22	0.21	0.2	0.28	0.3	0.36	0.36
Anhui	0.12	0.14	0.16	0.17	0.17	0.19	0.18	0.16	0.2	0.2	0.24	0.29
Fujian	0.18	0.17	0.16	0.17	0.17	0.2	0.19	0.17	0.24	0.24	0.32	0.32
Jiangxi	0.13	0.14	0.16	0.16	0.19	0.19	0.18	0.14	0.18	0.18	0.23	0.22
Shandong	0.2	0.19	0.22	0.27	0.24	0.26	0.29	0.24	0.23	0.24	0.32	0.31
Henan	0.14	0.14	0.17	0.18	0.15	0.18	0.16	0.12	0.19	0.16	0.2	0.2
Hubei	0.15	0.16	0.17	0.18	0.2	0.21	0.17	0.15	0.21	0.21	0.24	0.34
Hunan	0.16	0.18	0.18	0.2	0.21	0.24	0.22	0.27	0.21	0.2	0.24	0.24
Guangdong	0.21	0.22	0.24	0.24	0.25	0.26	0.24	0.25	0.29	0.29	0.38	0.34
Guangxi	0.11	0.11	0.14	0.14	0.15	0.19	0.18	0.17	0.22	0.23	0.24	0.21
Hainan	0.1	0.11	0.12	0.13	0.15	0.14	0.15	0.12	0.17	0.18	0.18	0.19
Chongqing	0.13	0.13	0.17	0.19	0.18	0.18	0.19	0.13	0.2	0.19	0.24	0.22
Sichuan	0.15	0.15	0.19	0.2	0.21	0.2	0.22	0.18	0.2	0.21	0.23	0.23
Guizhou	0.09	0.11	0.12	0.13	0.13	0.15	0.17	0.1	0.13	0.14	0.16	0.15
Yunnan	0.12	0.15	0.16	0.18	0.16	0.18	0.15	0.14	0.19	0.21	0.22	0.2
Shaanxi	0.13	0.12	0.16	0.15	0.2	0.22	0.22	0.19	0.21	0.2	0.22	0.22
Gansu	0.12	0.14	0.15	0.15	0.18	0.18	0.17	0.17	0.17	0.17	0.18	0.2
Qinghai	0.12	0.13	0.16	0.16	0.17	0.22	0.21	0.16	0.21	0.22	0.25	0.23
Ningxia	0.1	0.12	0.14	0.14	0.14	0.16	0.14	0.12	0.16	0.19	0.17	0.19
Xinjiang	0.12	0.13	0.16	0.16	0.17	0.21	0.19	0.16	0.21	0.2	0.23	0.23

**Table 7 ijerph-19-06994-t007:** Unit root test results.

Variable	LLC Inspection	IPS Inspection	ADF Inspection	PP Inspection	Conclusion
CO	−2.85 ***	0.48	−5.04	−5.17	unstable
ΔCO	−4.81 ***	−4.81 ***	6.83 ***	8.89 ***	smooth
GE	−5.83 ***	−3.94 ***	3.74 ***	1.52	smooth
ΔGE	−7.17 ***	−8.17 **	5.88 **	36.64 ***	smooth

Note: *** means *p* < 0.001; ** means *p* < 0.01.

**Table 8 ijerph-19-06994-t008:** Co-integration test results.

Test Method	Test Statistics	Test Result	Conclusion
Pedroni	Panelv-Statistic	−2.42	Co-integration correlation
	Panelrho-Statistic	0.56	
	PanelPP-Statistic	−6.31 ***	
	PanelADF-Statistic	−2.58 ***	
	Grouprho-Statistic	1.72 ***	
	GroupPP-Statistic	−7.28	
	GroupADF-Statistic	−2.02 **	
Johansen	None	177.7 ***	Co-integration correlation
	Atmost1	276.4 ***	

Note: *** means *p* < 0.001; ** means *p* < 0.01.

**Table 9 ijerph-19-06994-t009:** Test results of optimal lag order.

Lag	AIC	BIC	HQIC
1	−4.83	−4.04	4.51
2	−5.13 *	−4.23 *	−4.77 *

Note: * means *p* < 0.05.

**Table 10 ijerph-19-06994-t010:** Granger causality test results.

Null Hypothesis	Observations	df	chi2 Statistics	Conclusion
GE is not the cause of L_CO	270	2	7.21 ***	Reject the null hypothesis
CO is not the cause of L_GE	270	2	32.33 ***	Reject the null hypothesis

Note: *** means *p* < 0.001.

## Data Availability

Data available in a publicly accessible repository that does not issue DOIs. Publicly available datasets were analyzed in this study.This data can be found here: https://navi.cnki.net/knavi/yearbooks/YINFN/detail?uniplatform=NZKPT (accessed on 4 April 2022).

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
