# Peer review of "The Impact of Coordinated Development of Ecological Environment and Technological Innovation on Green Economy: Evidence from China"

_ijerph, 2022, doi:10.3390/ijerph19126994_

Round 1
Reviewer 1 Report
Dear authors,
I wish you all the best in your work.
It was a pleasure to read your work, highlighting how the coordination degree of ecological environment and technological innovation affects the green economy. This issue is very pertinent in the current context. The results are aligned with the objectives, the methods were employed appropriately.
Well done, congratulations.
However, allow me to give some suggestions to improve your work, such as:
Literature review
I suggest improving the structure of the literature review. This section has very long paragraphs; the authors may include subsections or include bullets. It would also be important to reinforce your work's additional contributions/originality compared to the work developed by Zhao J (2021).
Research Methods and Data Sources
The authors fail to explain the logic of the methodology and methods applied, such as:
- Better explain the choice of the 2005-2016 period. Do these results still make sense in the current 2022 context?
- Better explain the research design, approach and the procedures for planning it
- Being secondary data, the citations of the sources of the databases are missing.
A strong point of the work is the diversity of data analysis methods, but I would suggest a better explanation of the methodological choices.
- Results and discussion
Well-structured and coherent results
In my opinion, the Discussion section is missing. I would read more about possible practical utilization of your results (e.g. within implications or as a separate section within discussion). The conclusion just repeats what was had described in the results section.
For example, the authors set, "Among them, the comprehensive level of the green economy is the highest, the development level of technological innovation is the second, and the comprehensive level of ecological environment is the lowest." The authors should further discuss this result, such as justification, factors, drivers, and context of economic development in this period.
I hope to help you improve your paper so the journal can publish it.
Other than that, well done!
Author Response
Thankyou for your sugesstions on this paper,we have modified this paper accurding to your sugesstions, please see the attachment.

Reviewer 2 Report
I have a number of issues with this paper.
Line 34 - I have no idea what 'the reform' refers to
Line 37 does not clearly state what statistics are referred to
I feel that from line 51 there is limited discussion of what the empirical research actually is. Please explain what it is better.
Between line 51 - 56 this is one sentence. This is not acceptable English. This needs breaking up into at least 2 sentences and in the process provide greater clarity of meaning.
From line 57 I am not clear on the nature of the literature review and its meaning. What is being reviewed and how is it being reviewed. This needs far greater clarity.
In line 58 the research methods and data sources require greater connection to which part and what discussion.
In line 57 - 59 requires greater delineation and explanation. I am not clear on what is being discussed.
Line 63 - Reference to 'shit et al.' may need attention and adjustment
Between lines 65- 68 discussion may need clarification and further work on what is meant here.
There is a problem in my view on the discussion of literature. on line 68 - 70 it is not clear to me what the discussion on Zhu et al. means. This seems to me to be important. What is being discussed here. This is representative of other discussion of research.
The discussion between lines 61 - 109 require different paragraphs. in parts it is incoherent. The reference to 'it' on line 102 is by no means clear.
The discussion on lines 103 - 105 is totally obscure to me.
Conclusions drawn from literature review on lines 110 - 129 are not clear from earlier discussion.
Between line 118 - 129 this is all one sentence. Apart from being totally unclear as to meaning it is totally unacceptable as English. This needs rewriting.
Full stop at end of line 133.
The discussion in Part 3 is something I am not familiar with. I am fully prepared to accept the expertise of the writer who is clearly more mathematically orientated than writing in narrative English. The author is clearly very experienced in writing complex formulae.I am not experienced in this area.
Line 233 - 4 is not a complete sentence and is missing a full stop.
A major criticism is that the author is not defining nor explaining exactly is meant by comprehensive development level, economic environmental technology and green economy.please - this requires clear definition and explanation on how this is applied in this article. i do not feel this has been given at all. Similarly the various test done - which I presume have been completed properly - are not fully explained. As a result I totally reject the acceptability of the conclusion, its meaning and its ability to support the results of earlier test.
Author Response
Thankyou for your sugesstions ,we have modified the papr according to your sugesstions ,please see the attcahment.

Round 2
Reviewer 2 Report
I would prefer a fuller explanation in comment 18 but otherwise you have responded comprehensively to matters raised. Thank you.